# Interplay between an ATP-binding cassette F protein and the ribosome from *Mycobacterium tuberculosis*

Zhicheng Cui [1], Xiaojun Li[1], Joonyoung Shin [1], Howard Gamper [2], Ya-Ming Hou [2], James C. Sacchettini[1] & Junjie Zhang [1]✉

EttA, energy-dependent translational throttle A, is a ribosomal factor that gates ribosome entry into the translation elongation cycle. A detailed understanding of its mechanism of action is limited due to the lack of high-resolution structures along its ATPase cycle. Here we present the cryo-electron microscopy (cryo-EM) structures of EttA from *Mycobacterium tuberculosis (Mtb)*, referred to as MtbEttA, in complex with the *Mtb* 70S ribosome initiation complex (70SIC) at the pre-hydrolysis (ADPNP) and transition (ADP-VO$_4$) states, and the crystal structure of MtbEttA alone in the post-hydrolysis (ADP) state. We observe that MtbEttA binds the E-site of the *Mtb* 70SIC, remodeling the P-site tRNA and the ribosomal intersubunit bridge B7a during the ribosomal ratcheting. In return, the rotation of the 30S causes conformational changes in MtbEttA, forcing the two nucleotide-binding sites (NBSs) to alternate to engage each ADPNP in the pre-hydrolysis states, followed by complete engagements of both ADP-VO$_4$ molecules in the ATP-hydrolysis transition states. In the post-hydrolysis state, the conserved ATP-hydrolysis motifs of MtbEttA dissociate from both ADP molecules, leaving two nucleotide-binding domains (NBDs) in an open conformation. These structures reveal a dynamic interplay between MtbEttA and the *Mtb* ribosome, providing insights into the mechanism of translational regulation by EttA-like proteins.

[1] Department of Biochemistry and Biophysics, Texas A&M University, College Station, TX 77843, USA. [2] Department of Biochemistry and Molecular Biology, Thomas Jefferson University, Philadelphia, PA 19107, USA. ✉email: junjiez@tamu.edu

ATP-binding cassette F (ABC-F) proteins are wide-spread among bacteria and eukaryotes, with dozens of distinct groups[1–3]. Although belonging to the universal ABC transporter superfamily, they are unlikely to function as transporters due to the lack of transmembrane domains[4,5]. Instead, a variety of ABC-F proteins are involved in different aspects of protein synthesis. An early example is the eukaryotic Elongation Factor 3 (eEF3), a member of ABC-F proteins, which has potential roles in E-site tRNA release and ribosome recycling[6,7]. Later, more ribosome-associated ABC-F proteins were discovered in eukaryotes, many of which have been proposed to be involved in response to stress, translation initiation, and ribosome biogenesis[8–11]. In bacteria, the first structurally determined ABC-F protein, EttA from *Escherichia coli* (referred to as EcoEttA), was found to bind to the ribosome and "throttle" (or regulate) its entry into the translation elongation cycle, depending on the ATP/ADP ratio[12,13]. Interestingly, a subset of bacterial ABC-F proteins has been identified to protect the ribosome against clinical antibiotics and classified into different groups based on the types of antibiotics they protect against[14,15]. In recent years, direct protection of ribosome against antibiotics by these antibiotic resistance (ARE) ABC-F proteins has stepped into the limelight, which is supported by biochemical and structural studies of various AREs, disfavoring the previous idea that they function as efflux pumps[16–18].

ABC-Fs are generally composed of two tandem nucleotide-binding domains (NBDs) connected by a linker of ~60–100 amino acids[2]. In spite of various lengths in the linker region, distinct features among ABC-F proteins include an extra N-terminal domain, an "Arm" domain insertion within the first NBD (NBD1), and a C-terminal extension[1]. Like other members in the ABC superfamily, most ABC-F proteins are ATPases with highly conserved ATP binding/hydrolysis motifs, albeit with variations of the aromatic residue at the A-loop within NBD1[12]. The two NBDs interact with each other in a head-to-toe fashion, sandwiching two ATP molecules[19]. Movement between the two NBDs during ATP hydrolysis, referred to as a "clamping" between the open and closed conformations, is proposed based on available structures of different ABC-F proteins at different nucleotide-bound states[12,13,16,17,20].

Structural investigation into the mechanism of action of ABC-F proteins has primarily been focused on AREs, such as MsrE (resistance to macrolides, streptogramin B, ketolides) and VmlR (resistance to virginiamycin M and lincomycin)[16,17]. The mechanism of conferring antibiotic resistance by MsrE is proposed as direct dispersions of each antibiotic from the ribosome, due to steric clashes between the tip of the linker and each antibiotic[16]. In addition, the bound MsrE can induce conformational changes in the acceptor stem of the P-site tRNA, potentially destabilizing the bound antibiotics from the active sites of the ribosome[2]. Notably, structural and mutational studies of VmlR also provided evidence of an alternative mechanism for the dissociation of antibiotics, in which VmlR caused conformational changes around the peptidyl transferase center (PTC) in the 23S ribosomal RNA (rRNA)[17]. Another focus was the characterization of EcoEttA, which was not involved in conferring antibiotic resistance, probably due to its shorter linker compared to AREs. The only available structures of EttA-like ABC-F proteins are a crystal structure of nucleotide-free EcoEttA alone and a low-resolution cryo-EM structure of ATP-bound EcoEttA in complex with the *E. coli* ribosome[12,13]. These studies show that EcoEttA binds at the ribosome E-site and interacts with the initiator tRNA to regulate the ribosome entry into the elongation cycle. However, high-resolution structures of EttA-like proteins at different states during ATP hydrolysis and their interactions with the ribosome are needed to better understand the mechanism, such as the cooperativity between the two NBDs and the interplay between the EttA and the ribosome.

Rv2477c from *Mycobacterium tuberculosis* (*Mtb*) is an EttA-like ABC-F protein[21], which was previously mistaken as an efflux pump[22]. Unlike EcoEttA, *Mtb* Rv2477c (termed as MtbEttA from now on) is responsive to antibiotic treatments[23] and essential for cell growth based on a study by saturating transposon mutagenesis[24] (https://mycobrowser.epfl.ch/genes/Rv2477c). MtbEttA could be a drug target in *Mtb*, which causes Tuberculosis. This disease leads to ~2 million deaths worldwide annually[25]. In our in vitro translation assays (Supplementary Fig. 1), we showed that, although MtbEttA was unable to protect the ribosome from erythromycin, it increased the translation activity of the ribosome at low erythromycin concentrations. In order to better understand how MtbEttA enhances translation activity of the ribosome, we performed structural analyses of the *Mtb* ribosome-bound MtbEttA along the trajectory of ATP hydrolysis. We determined high-resolution cryo-EM structures of the ribosome-bound MtbEttA at the pre-hydrolysis (ADPNP) and transition (ADP-VO$_4$) states. Different degrees of ribosomal intersubunit rotation are observed between the pre-hydrolysis and the transition states. In the pre-hydrolysis state, we observed an asymmetric engagement of conserved ATP-hydrolysis motifs around the two nucleotide-binding sites (NBSs) in MtbEttA and an increased flexibility around the PTC of the *Mtb* ribosome, both of which are correlated with the ribosomal intersubunit rotation. We also solved a crystal structure of MtbEttA alone in the post-hydrolysis (ADP) state with two MtbEttA forming a domain-swapped dimer. The two NBDs, which form the two NBSs, are apart from each other. They are in an "open" conformation, therefore cannot tightly engage the nucleotides. These structures reveal a dynamic interplay between the ribosome and MtbEttA during the course of ATP hydrolysis and provide insights into the mechanism of EttA-regulated translation.

## Results

**Cryo-EM structures of *Mtb* ribosome in complex with MtbEttA at the pre-hydrolysis (ADPNP) and transition (ADP-VO$_4$) states.** To obtain a stable complex between the *Mtb* ribosome and MtbEttA, we assessed multiple conditions by adding mRNA, tRNA, and different nucleotide analogs (Supplementary Fig. 2). Consistent with previous studies[17], a P-site tRNA and a non-hydrolyzable ATP analog ADPNP were necessary to form a stable MtbEttA-ribosome complex. In addition, we discovered that MtbEttA can still bind to the 70S in the presence of ADP-VO$_4$, an analog mimicking the transition state of ATP hydrolysis. We then reconstituted an *Mtb* 70S initiation complex (70SIC containing the mRNA and initiator tRNA fMet-tRNA$^{fMet}$) bound to MtbEttA with ADPNP or ADP-VO$_4$ in vitro, respectively, for our structural studies.

For the *Mtb* 70SIC-bound MtbEttA in the pre-hydrolysis state, single-particle cryo-EM 3D classification and refinement yielded two distinct subpopulations, named Pre_R0 and Pre_R1, to the resolutions of 2.97 and 3.23 Å, respectively (Supplementary Fig. 3). In addition, other subpopulations, including the classic nonrotated 70S with P/P tRNA, fully rotated 70S with P/E tRNA, nonrotated 70S with P/P and E/E tRNAs, and 50S, were also resolved to 2.76, 2.8, 2.71, and 3.0 Å resolutions, respectively. The core region of the 70S with P/P and E/E tRNAs is at an even higher resolution, which enabled us to identify several conserved RNA modifications in the 23S and 16S rRNAs (Supplementary Fig. 4).

Similar to the pre-hydrolysis state, two distinct subpopulations were revealed for the *Mtb* 70SIC-bound MtbEttA in the ATP-hydrolysis transition state, namely Trans_R0 and Trans_R1, to the resolutions of 2.79 and 3.1 Å, respectively (Supplementary Fig. 5). The dataset also included subpopulations without

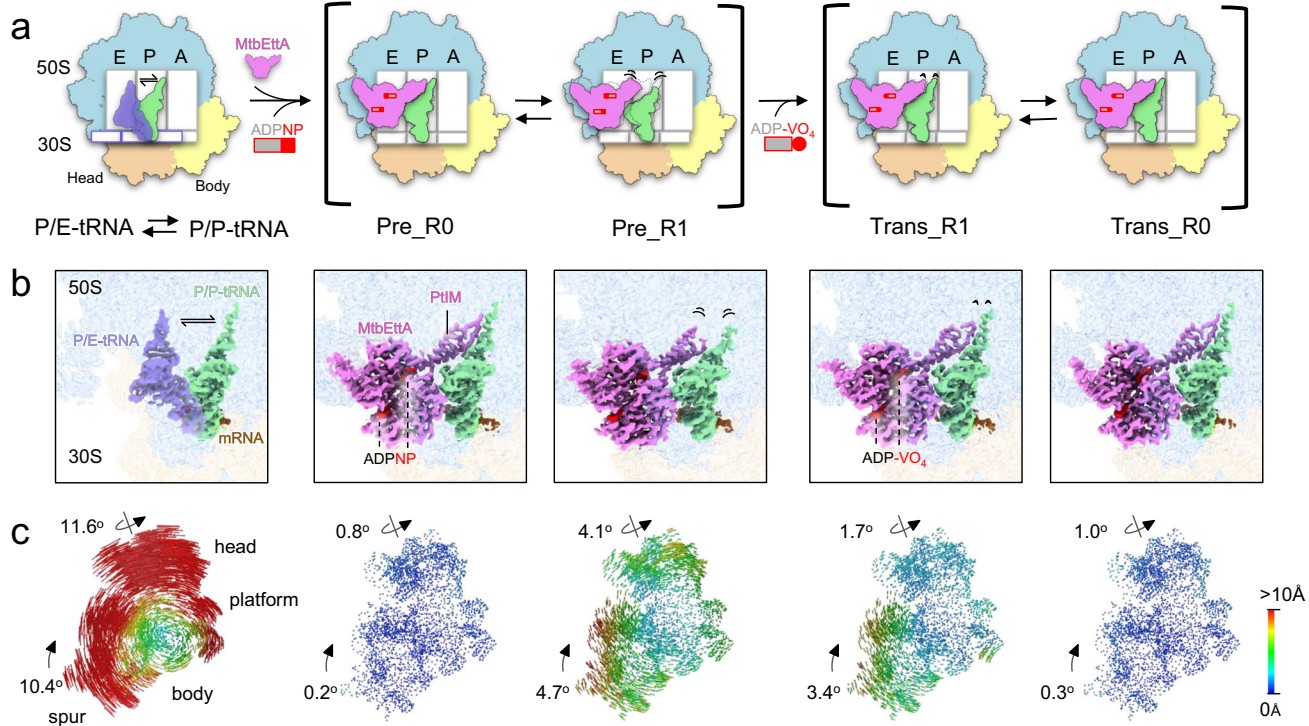

**Fig. 1 Cryo-EM structures of *Mtb* 70SIC in complex with MtbEttA in the pre-hydrolysis and transition states. a** Cartoon representations of *Mtb* ribosome in different states, labeled with the E, P, and A sites, as well as the 50S and domains (head and body) of the 30S. The first column shows the transition of tRNA between the P/P (light green) and P/E (transparent medium blue) positions. The second and third columns show that MtbEttA (violet), in the Pre_R0 and Pre_R1 states, respectively, binds at the E-site of *Mtb* 70SIC and interacts with the P-site tRNA. The fourth and fifth columns show the corresponding 70SIC-MtbEttA complexes in the Trans_R1 and Trans_R0 states, respectively. Dashed outlines at the E and P sites (third column) show the position of MtbEttA and the tRNA in the Pre_R0 state. The length of the black curved lines indicates the flexibility in these regions. **b** Cryo-EM maps showing the density of tRNAs and MtbEttA with the 50S (light blue) and the 30S (light yellow) transparent in the background. PtIM and ATP analogs are labeled. The color scheme follows panel (**a**). **c** Analysis of 30S rotations in the P/E-tRNA or MtbEttA-bound states, relative to the non-rotated P/P-tRNA state. The head, body, platform, and spur of the 30S subunit are labeled. Rainbow-colored vectors represent the displacements between equivalent C4′ or Cα atoms of the 30S subunits, with red and blue indicating large and small movements, respectively.

MtbEttA bound, such as the 70S with P/P tRNA, with P/E tRNA, with both P/P and E/E tRNAs and the 50S alone, which we did not further process due to the similarity to the corresponding subpopulations in the pre-hydrolysis state.

MtbEttA binds at the E site of the *Mtb* 70SIC and interacts with several ribosomal proteins and RNAs, as well as the P-site tRNA. The linker region, termed as the P-site tRNA interaction motif (PtIM)[12], extensively interacts with the P-site tRNA and the 23S rRNA (Figs. 1a, b, 2b). The overall structural difference of the *Mtb* ribosome among these states is the degree of intersubunit motion between the 30S and the 50S, which can be measured after aligning them based on the 23S rRNA. In the two pre-hydrolysis states, the Pre_R0 30S has a subtle body rotation of ~0.2° and head swivel of ~0.8° relative to the classic nonrotated P/P-tRNA state, while the Pre_R1 30S has a body rotation and head swivel of ~4.7° and ~4.1°, respectively (Fig. 1c, 2nd and 3rd columns). In the transition states, the Trans_R1 30S shows a body rotation of ~3.4° and head swivel of ~1.7°. The Trans_R0 state shares similar motion as the Pre_R0 state, with a body rotation of ~0.3° and head swivel of ~1.0°. Nevertheless, the rotational movement of the 30S in the presence of MtbEttA both in the pre-hydrolysis and transition states is modest compared to the fully rotated state as shown in the 70S with P/E tRNA, which displays a body rotation and head swivel of ~10.4° and ~11.6°, respectively (Fig. 1c, 1st column). These results indicate that binding of MtbEttA at the E site of the *Mtb* 70SIC reduces the overall movement of the *Mtb* 30S relative to the 50S.

In the Pre_R0 and Trans_R0 states when the 30S has a small movement, the tip of PtIM extends towards the PTC and shows stable interactions with the P-site tRNA (Fig. 1b, 2nd and 5th columns). Notably, PtIM and the P-site tRNA increase their flexibility as the head of the 30S swivels more. Particularly, the tip of PtIM and the CCA tail of the P-site tRNA display the largest flexibility in the Pre_R1 state, as the cryo-EM densities in these regions are weak (Fig. 1b, 3rd column). Even though the body rotation in the Trans_R1 state is comparable to that in the Pre_R1 state, only subtle variations of PtIM and tRNA are detected, presumably due to the smaller head swivel in the 30S. These observations demonstrate that the swiveling of the 30S head domain correlates with the displacements of PtIM of MtbEttA and the CCA tail of the P-site tRNA.

**Interactions between MtbEttA and the *Mtb* ribosome**. MtbEttA has a conserved ABC-F architecture with two tandem NBDs, an arm domain, PtIM, and a basic C-terminal tail (Fig. 2a). Using the Pre_R0 state as an example, the arm domain of MtbEttA interacts with the L1 stalk of the 50S (Fig. 2b). Surprisingly, contrary to the low-resolution cryo-EM structure of EcoEttA bound to an *E. coli* ribosome, the basic C-terminal tail of MtbEttA is inserted between the L1 stalk of the *Mtb* ribosome and the NBD1 of the MtbEttA (Fig. 2c). The C-terminal polypeptide backbone of MtbEttA is clearly resolved (Supplementary Fig. 4c). When bound to the ribosome, PtIM of MtbEttA is more extended than the one in EcoEttA (Supplementary Fig. 6a). The first α-helix of PtIM (PtIM α1), which connects to the NBD1, has extensive interactions with Helix 68 of the 23S rRNA (Fig. 2d). Moreover, the positively-charged tip of PtIM is stretched and inserted into

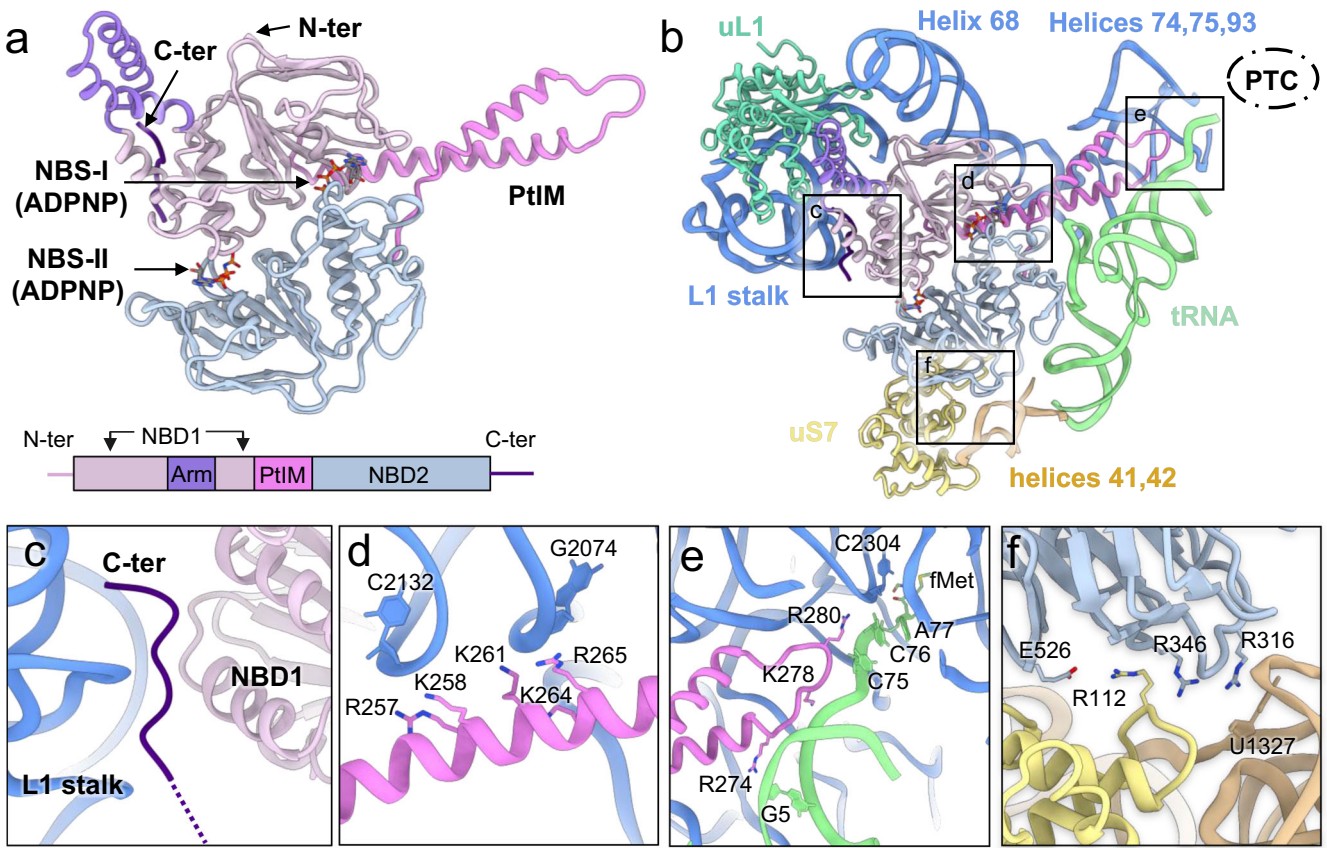

**Fig. 2 Structure of MtbEttA and its interaction with the *Mtb* 70SIC in the Pre_R0 state. a** The structure and domain composition of MtbEttA. Nucleotide-binding domains 1 (NBD1, residues 2-93 and 137-240) and 2 (NBD2, residues 308-545), arm domain (residues 94-136), P-tRNA interacting motif (PtIM, residues 239-307), and the C-terminal basic tail (residues 550-557) are colored in thistle, light steel blue, medium purple, violet, and indigo, respectively. Nucleotide-binding sites (NBS-I and NBS-II) are indicated by arrows. **b** The overall interactions between MtbEttA and surrounding RNAs and proteins. The relative location of the peptidyl transferase center (PTC) is outlined by a dashed oval. Structural details of boxed regions are described in panels (**c**–**f**). **c** The C-terminal tail is inserted between the L1 stalk and NBD1. **d** PtIM α1 is in close contact with Helix 68 of 23S rRNA. **e** The tip of PtIM reaches towards the PTC, and interacts with the CCA tail of the P-site tRNA, Helices 74, 75, and 93 of the 23S rRNA. **f** Interactions between MtbEttA and the 30S. A salt bridge is observed between E526 (MtbEttA) and R112 (uS7). Basic residues in MtbEttA (R316 and R346) point towards helices 41 and 42 of the 16S rRNA. Several rRNA nucleotides are labeled to show the regions of interactions.

the region surrounded by Helices 74, 75, 93 of the 23S rRNA and the acceptor arm of the P-site tRNA (Fig. 2b, e). The electrostatic potential of MtbEttA confirms that its positively charged surface faces the negatively charged phosphate RNA backbone (Supplementary Fig. 7). Notably, the sidechain of R280, at the tip of PtIM, points towards the PTC in a fashion that is parallel with the backbone of the CCA tail of the P-site tRNA (Fig. 2e). Interestingly, this arginine residue is conserved among different EttA-like proteins (Supplementary Fig. 8), suggesting a functional role in its interaction with the neighboring RNA backbones. Interactions between MtbEttA and the 30S involve a salt bridge between E526 of NBD2 and R112 of the uS7 protein, as well as electrostatic interactions between the basic residues R316 and R346 of NBD2 and the RNA backbones of helices 41 and 42 in the 16S (Fig. 2f). In summary, MtbEttA binds the *Mtb* 70SIC via a network of charge-charge interactions with the ribosome and the P-site tRNA; and these interactions are conserved in our structures of MtbEttA in complex with the *Mtb* 70SIC, except for the tip of PtIM in the Pre_R1 state when it becomes flexible.

**Remodeling of the P-site tRNA and the ribosomal intersubunit bridge B7a by MtbEttA as the 30S moves.** The intersubunit movement within the 70S is often accompanied by a shift of the

P-site tRNA. In addition, different ribosomal factors can result in various intermediate positions of the P-site tRNA[26–28]. Previous structures of ribosome in complex with MsrE and VmlR revealed a novel P-site tRNA conformation, in which the acceptor stem was displaced by the tip of PtIM and redirected from the PTC toward the A site[16,17]. PtIM from MtbEttA is unlikely to cause such a drastic displacement due to the shorter PtIM tip (Supplementary Fig. 6b, c, f, g). Indeed, we find that the acceptor stem of the P-site tRNA in the Pre_R0 and Trans_R0 states remains in the same position as the classic P/P-tRNA state without the MtbEttA. However, the elbow region of P-tRNA tilts ~6° toward the E site, presumably due to the interactions with the bound MtbEttA (Supplementary Fig. 6d). Furthermore, the elbow moves ~16° toward the E site in the Pre_R1 state, which is comparable to that in ARE-ribosome complexes (Supplementary Fig. 6e). Nevertheless, the CCA tail and acceptor stem remain at the P-site but with increased flexibility.

Compared to the Pre_R0 state, the MtbEttA in the Pre_R1 state shifts along the same direction as the 30S for ~6 Å (Fig. 3a, b). Notably, the 50S protein bL31, which forms the ribosomal intersubunit bridge 1b (B1b), along with protein uL5, moves ~2 Å opposite to the direction of the 30S head swiveling (Fig. 3a, b). In addition, the 5S rRNA and Helix 84 of the 23S rRNA show an upward shift (Supplementary Fig. 9a). These movements around

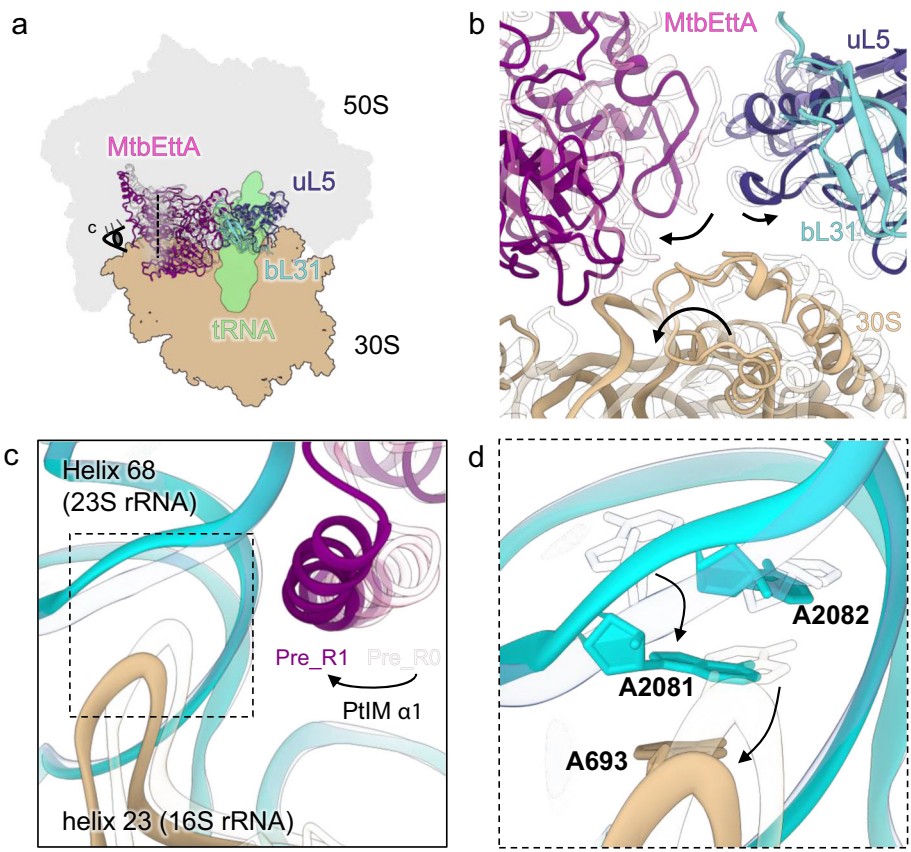

**Fig. 3 Remodeling of the *Mtb* 70SIC upon binding MtbEttA in the Pre_R1 state. a** Models of the 70SIC-MtbEttA complex in the Pre_R0 (transparent) and Pre_R1 (solid color) states are superimposed based on the 23S rRNA. 50S, 30S, and tRNA for the Pre_R0 state are shaded in the background. MtbEttA, bL31, and uL5 are shown as models. **b** A zoom-in view at the split interface. Directions of movements for MtbEttA, 30S, bL31, and uL5 are indicated by black arrows. **c** Rearranged 50S-30S interface in the Pre_R1 state. Direction and position of the view are indicated by the eye cartoon and a dashed vertical line in (**a**). **d** A close-up view of the rearranged base stacking in the Pre_R1 state is shown, with arrows indicating movements of corresponding nucleotides from the Pre_R0 state (transparent).

the central protuberance (CP) in the Pre_R1 state result in a split at the interface between the two ribosomal subunits (Fig. 3b; Supplementary Movie 1). Although this is also observed in the fully-rotated 70S with the P/E tRNA, the CP does not move to the same extent as in the Pre_R1 state (Supplementary Fig. 9b). Such a movement of the CP is likely due to both the occupancy of MtbEttA at the E-site and the rotation of the 30S. In the transition states when the 30S head barely swivels, bL31 and uL5 do not show major movements (Supplementary Fig. 9c). In addition, in the Pre_R1 state, we observe the 23S rRNA Helix 89 and protein uL16 around the PTC becomes more flexible (Supplementary Fig. 10). Molecular dynamics simulation has suggested that, during the tRNA translocation from A/T to A/A sites, displacement of Helix 89 of the 23S rRNA is necessary to avoid steric clashes with the elbow of the tRNA[29]. Therefore, increased flexibility of Helix 89 around the PTC, observed in the Pre_R1 state, may prime the ribosome to accommodate the A-site tRNA. However, whether MtbEttA interacts with the 70SIC before or after the binding of the EF-Tu-aa-tRNA needs to be investigated.

Furthermore, the intersubunit bridge B7a between the 50S and the platform region of the 30S is rearranged in the Pre_R1 state (Fig. 3d). Particularly in the Pre_R0 state, there is a base stacking between nucleotides A693 from the 16S rRNA and A2082 from the 23S rRNA. However, in the Pre_R1 state, movements of the 30S and PtIM α1 cause a separation of A693 from A2080, allowing nucleotide A2081 of the 23S rRNA to flip and insert between them to form new base stackings. These newly formed

base stackings in the Pre_R1 state were not observed in any other ribosomal structures and may contribute to the stability of the specific intermediate 30S rotation in the Pre_R1 state (Supplementary Fig. 11).

**Structural variations of ribosome-bound MtbEttA in the pre-hydrolysis and transition states**. The ATPase activity is essential for the function of ABC-F proteins[12,18]. Moreover, the ribosome significantly increases the ATP hydrolysis activity of EcoEttA in vitro[12]. With these high-resolution cryo-EM structures at pre-hydrolysis and transition states, we can quantify such structural variations of MtbEttA along the reaction trajectory of the ATP hydrolysis in the presence of the ribosome.

Conserved motifs of ABC ATPases are present in MtbEttA at both NBSs. These motifs include the Walker A, Walker B, H-switch, Q-loop, A-loop, and signature motifs (Supplementary Fig. 12a). When complexed with the ribosome at both the pre-hydrolysis and transition states, MtbEttA is in a "closed" conformation, in which the nucleotides along with the corresponding $Mg^{2+}$ ions are clearly visualized and engaged by the conserved ATP-binding motifs of the two NBSs (Supplementary Fig. 12b).

MtbEttA exhibits significant structural variations in different nucleotide states when bound to the *Mtb* ribosome. The most prominent conformational difference is observed between the Pre_R0 and Pre_R1 states (Fig. 4a). Compared to the Pre_R0 state, the tip region of PtIM in the Pre_R1 state shows a more relaxed

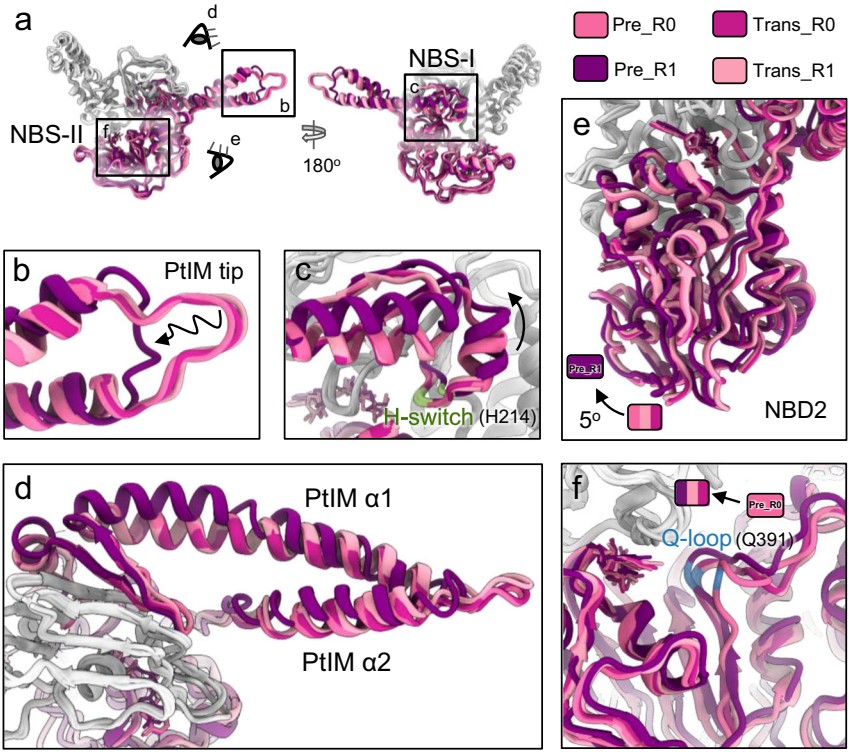

**Fig. 4 Structural plasticity of MtbEttA in the pre-hydrolysis and transition states. a** Overall structural comparison of MtbEttA in the pre-hydrolysis and transition states. Models are aligned based on NBD1. NBD1 and arm domain are colored in light gray. PtIM and NBD2 are colored from light pink to dark purple for different states. The close-up views of the tip of PtIM and the base of PtIM α1 are shown in (**b** and **c**), respectively. Residue H214 of the H-switch in NBD1 is colored in pale green. **d** Top view of MtbEttA from the direction of the eye cartoon in **a**. Focused views of NBD2 and the Q-loop in NBD2 are shown in (**e** and **f**), respectively. The curved arrow indicates the rotation of NBD2 in (**e**), and Residue Q391 is colored in dodger blue in (**f**). Nucleotides are shown as stick models in (**c**, **e**, and **f**).

conformation (Fig. 4b), with Cα distances of corresponding residues at a range of 5–10 Å, while it shows limited changes in the ATP-hydrolysis transition states (Supplementary Fig. 13). PtIM α1 also deforms into a bent α-helix (Fig. 4d), which shifts the H-switch loop at NBS-I (labeled green in Fig. 4c), dissociating it from the bound nucleotide (stick models in Fig. 4c). Furthermore, NBD2 of the Pre_R1 state moves ~5° more toward NBD1 (Fig. 4e), closing NBS-II to engage the Q-loop onto the nucleotide (Fig. 4f).

**Asymmetric engagements of ADPNPs in the pre-hydrolysis states.** The conformational changes in MtbEttA in the pre-hydrolysis states have led to asymmetric ADPNP engagements by the conserved ATP-binding motifs of the two NBSs (Supplementary Movie 2). In the Pre_R0 state when the 30S has little rotation, one ADPNP is fully enclosed by the conserved motifs at NBS-I, while the Q-loop at NBS-II is flexible and away from the other ADPNP (Fig. 5a, b). Subsequently in the Pre_R1 state, the residues H15 in the A-loop and H214 in the H-switch at NBS-I move outward for 2.5 and 3.5 Å, respectively (Fig. 5c). On the other hand, the residue Q391 in the Q-loop at NBS-II flipped inward for a distance of 4.7 Å to interact with ADPNP (Fig. 5d). By contrast in the ATP-hydrolysis transition states, all the conserved motifs in both NBSs are engaged with ADP-VO$_4$ (Fig. 5e–h).

**The crystal structure of ribosome-free MtbEttA at the post-hydrolysis (ADP) state.** Speculations were made by others that the two NBDs of ABC-F proteins could undergo conformational changes after ATP hydrolysis, as discovered in the ABC

superfamily[30]. However, the lack of structure of ABC-F proteins at post-hydrolysis state has limited our understanding of the potential conformational changes. Based on our result, MtbEttA in the ADP state did not form a stable complex with the ribosome (Supplementary Fig. 2). Therefore, we co-crystallized MtbEttA with ADP and determined its X-ray crystal structure at 2.9 Å resolution (Supplementary Table 1). In the crystal, the asymmetric unit of MtbEttA-ADP consists of a domain-swapped dimer, similar to that in the crystal structure of nucleotide-free EcoEttA (Supplementary Fig. 15a). Apparently, the conformations of both NBSs in the nucleotide-free EcoEttA are not compatible for stable nucleotide binding, as Pro16 and Tyr333 in the A-loop of NBS1 and NBS2, respectively, are either too far from or collide with the adenine ring of a potential nucleotide (Supplementary Fig. 15b, c). On the contrary, the corresponding residues, His15 and Tyr331, in the ADP-bound MtbEttA stack with the adenine ring of ADP, with the density of the ADP and magnesium ions clearly resolved in the crystal structure (Supplementary Fig. 15d, e). Notably, in solution, the population distribution of oligomeric states for EcoEttA and MtbEttA are different. For example, at a protein concentration of ~10 μM, EcoEttA has almost half of the population staying in the monomeric form[12], while MtbEttA is dominantly in the dimeric form (Supplementary Fig. 14a). Furthermore, the dimerization of the MtbEttA is significant in solution regardless of protein concentrations (Supplementary Fig. 16) or bound nucleotides (Fig. 6a).

In the crystal structure of the post-hydrolysis state of MtbEttA, one ADP molecule and a Mg$^{2+}$ ion are clearly resolved in each of the four NBSs in the domain-swapped dimer (Fig. 6b). Only subtle conformational differences are observed between the two halves in the crystal asymmetric unit (Supplementary Fig. 14b). In

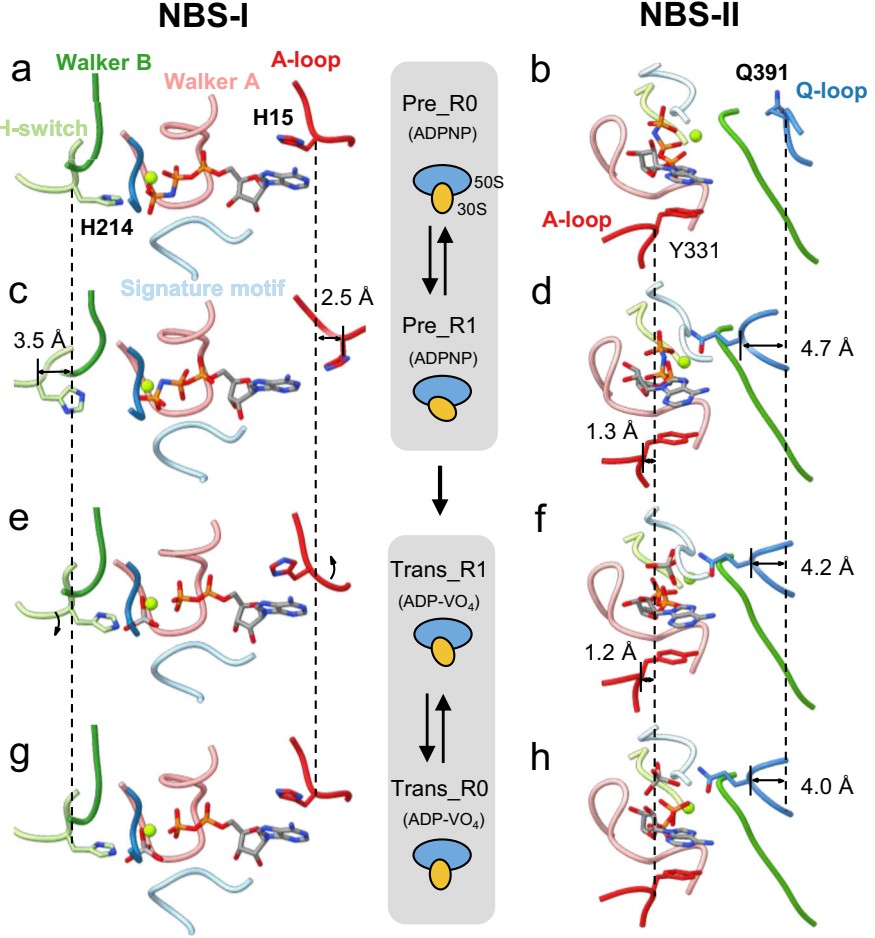

**Fig. 5 Asymmetric nucleotide engagements in the pre-hydrolysis states.** Nucleotide engagements in the NBSs for the Pre_R0 (**a** and **b**), Pre_R1 (**c** and **d**), Trans_R1 (**e** and **f**), and Trans_R0 (**g** and **h**) states. Different ATP-binding/hydrolysis motifs are colored and labeled as in panel (**a**). Magnesium is shown as a sphere and colored lime. Cartoons in-between the two columns indicate different states of the 70SIC-MtbEttA complex and different degrees of the 30S rotation. Dashed vertical lines aid the measurement of the shift of corresponding Cα atoms between different states.

order to understand the conformational change of MtbEttA at the post-hydrolysis state compared to the previous step of the ATP-hydrolysis (transition state), we focused on half of the domain-swapped dimer composed of the NBD1 from protomer A, and NBD2 from protomer B, which was analyzed in the same fashion for the crystal structure of the nucleotide-free EcoEttA[12].

To measure the conformational difference, we align the two structures of MtbEttA in the post-hydrolysis (ADP) state and the Trans_R0 state based on the NBD1. We observe a ~40° rotation of NBD2 away from NBD1 in the post-hydrolysis state (Fig. 6c, d), allowing the formation of two new salt bridges (E199/R499 and R216/E482, inset of Fig. 6d) between NBD1 and NBD2. These two salt bridges are absent in the Trans_R0 state (inset of Fig. 6c). Such an opening of the two NBDs is accompanied by the separation of the signature motif, Q-loop and H-switch, from the ADP, which shows a classic configuration in the ABC superfamily after ATP hydrolysis (Fig. 6e, f). In order to evaluate the effects of such a large domain movement of MtbEttA at the post-hydrolysis state, we superimposed the structure of the two NBDs of MtbEttA in the ADP state to the 70SIC-MtbEttA complex in the Trans_R0 state based on NBD1 (Supplementary Fig. 17; Supplementary Movie 3). Clear steric clashes are observed between the NBD2 and arm domain with uS7 and uL1, respectively. This may explain why the ADP-state MtbEttA cannot form a stable complex with the *Mtb* 70SIC and has to dissociate from the ribosome after ATP hydrolysis.

## Discussion

Using cryo-EM and X-ray crystallography, we have determined the structures of MtbEttA and its interaction with the *Mtb* ribosome in the course of ATP hydrolysis. Based on our results, we propose a general model for the mechanism of action of MtbEttA (Fig. 7): In the solution, MtbEttA stays at an equilibrium between monomers and dimers, with a major population as homo-dimers. In the dimeric form, MtbEttA is inactive since the dimer is too big to fit into the binding site in the *Mtb* ribosome. The ratio between the dimer and the monomer, in vitro, does not seem to be affected by protein concentrations (Supplementary Fig. 16) or nucleotide binding (Fig. 6a), but rather by high salt (e.g., NaCl) concentrations (Supplementary Fig. 14a), consistent with the fact that the dimer is stabilized by electrostatic interactions (Supplementary Fig. 14c). In *Mtb* cells, the physiological concentration of NaCl is ~250 mM[31], under which most of the MtbEttA will still exist as dimers. Upon dissociation of the two monomers, the two NBDs would undergo a large domain rearrangement to form a compact conformation—as the retention volume, on a size exclusion column for the monomeric MtbEttA, suggests it is in a globular form. The large population of the inactive dimers may serve as a reservoir of MtbEttA, which will dissociate into monomers to interact with the *Mtb* ribosomes. Such a hypothesis for the dimer/monomer switch needs to be tested. It is possible that the presence of the *Mtb* ribosome binding the monomeric MtbEttA drives the dimer/monomer

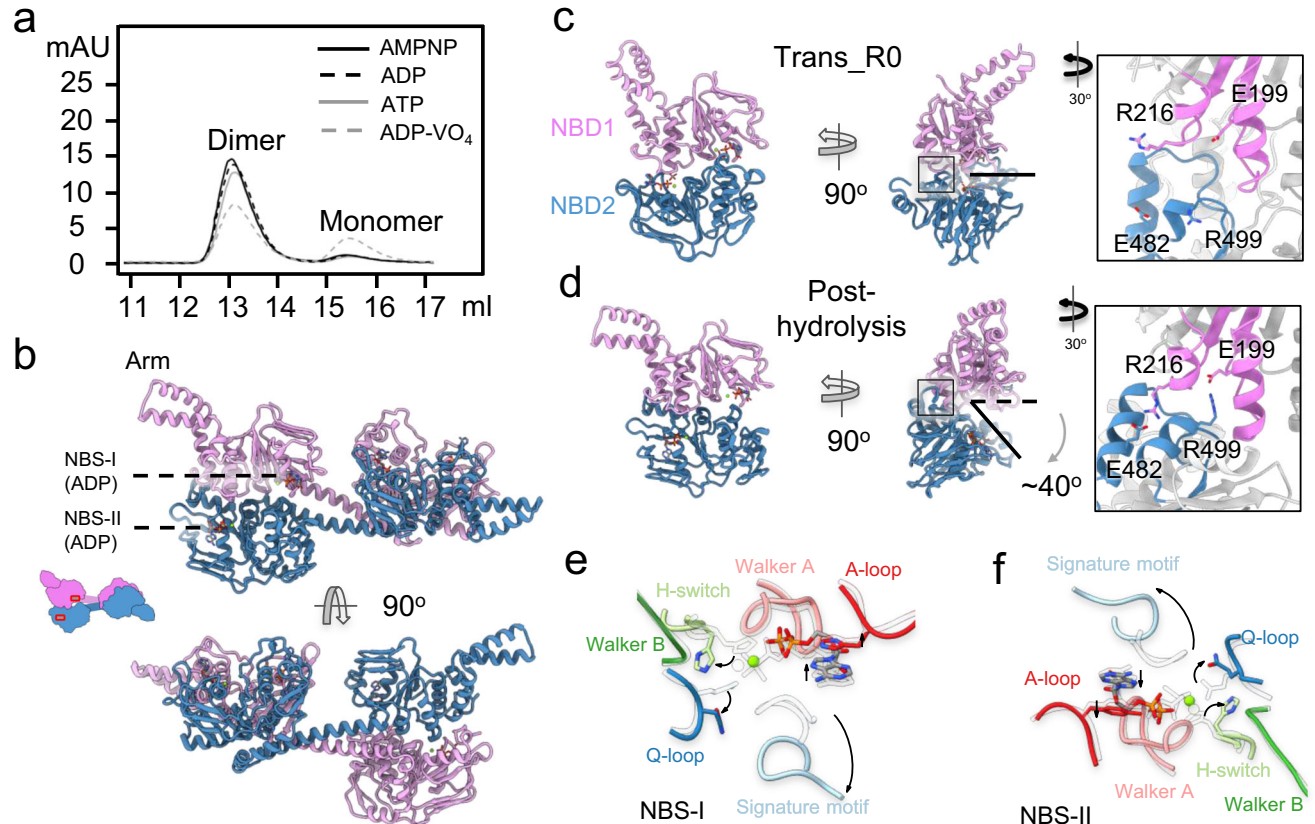

**Fig. 6 The crystal structure of MtbEttA in the ADP state. a** Size exclusion chromatography graph indicates the population distribution of dimer and monomer of MtbEttA with different nucleotides. **b** Crystal structure of MtbEttA bound with ADP. The protomer A (plum) and protomer B (steel blue) form a domain-swapped dimer. ADP molecules are resolved and labeled in both NBSs for each half of the domain-swapped dimer. Conformational differences between the two NBS-forming NBDs in the Trans_R0 (**c**) and post-hydrolysis (**d**) states. For the Trans_R0 state, the two NBDs are from the cryo-EM structure of the 70SIC-MtbEttA complex. For the post-hydrolysis state, NBD1 (residues 2–241) and NBD2 (306–550) are from Protomers A and B in the crystal structure, respectively, and form the two NBSs. The direction and degree of NBD2 opening relative to NBD1 are indicated in (**d**). Insets in **c** and **d** show zoom-in views of the interaction between two NBDs. **e**, **f** Movements of the conserved motifs in both NBSs from the Trans_R0 state (transparent) to the post-hydrolysis state (colored) are indicated by black curved arrows.

equilibrium towards the monomers. Admittedly, we cannot rule out that, in vivo, there exists a potential factor to regulate the oligomeric states of MtbEttA, which is yet to be discovered.

In the pre-hydrolysis state, the back-and-forth rotation of the *Mtb* 30S subunit is correlated with the alternating coupling and decoupling of ADPNP in the two NBSs. In the Pre_R0 state, in which the 30S is close to a classic nonrotated conformation, one ADPNP at the NBS-I is fully engaged by all the ATP-binding motifs of MtbEttA, while the other ADPNP at the NBS-II is disconnected from the Q-loop. As the 30S rotates to the Pre_R1 state, the ADPNP at the NBS-II is now fully engaged, but the ADPNP at the NBS-I is disconnected from the H-switch and the A-loop. The conformational changes of the H-switch and the A-loop at NBS-I are likely mediated by the bending of PtIM α1, which serves as a cantilever to transmit the effect of 30S rotation (Figs. 3c and 4d). In the pre-hydrolysis states, the rotation of the *Mtb* 30S facilitates the conformational changes of MtbEttA to fully engage the nucleotides in the two NBSs in a staggered way. By contrast in the transition states, nucleotides are always fully engaged by conserved ATP-hydrolysis motifs at both NBSs, presumably due to a smaller head swiveling of the 30S, which exerts less impact on the bound MtbEttA. On the other hand, ADP-VO₄ may stabilize the interaction between NBD1 and NBD2 within MtbEttA, locking MtbEttA in a compact monomeric form with both NBSs engaged onto the nucleotides, which

is consistent with an increased monomer population revealed in the size exclusion chromatography (Fig. 5a).

MtbEttA occupies the E-site on the *Mtb* 70SIC in both the pre-hydrolysis and transition states, reducing the movement of the 30S and stabilizing the tRNA in the P/P position, which functions similarly to the bacterial EF-P[32] and eukaryotic eIF5A[33]. Such a stabilization is favorable for the incorporation of the A-site tRNA, a rate-limiting step in peptide bond formation[34]. The role of the EttA-like proteins to facilitate the first peptide bond during translation initiation was previously proposed by a single-molecule fluorescence resonance energy transfer (smFRET) experiment on EcoEttA[13], and further visualized in our high-resolution cryo-EM structures of the MtbEttA in complex with the *Mtb* 70SIC. When the A-site tRNA binds the ribosome, the elbow and 3′-CCA tail of the tRNA navigate through a "corridor" to transit from the A/T to the A/A position[29], which requires the movement of Helix 89 of the 23S rRNA. Our structures show the movement of Helix 89 in the presence of MtbEttA in the Pre_R1 state, which may further increase the rates for a successful accommodation of the A-site tRNA. Moreover, the impact of MtbEttA in sampling different conformations of the 30S to facilitate translation is consistent with the notion that rotations of the 30S facilitate the A-site tRNA accommodation[35].

It is likely that the presence of the third phosphate in the ATP molecule allows the engagement of the signature motif, at least

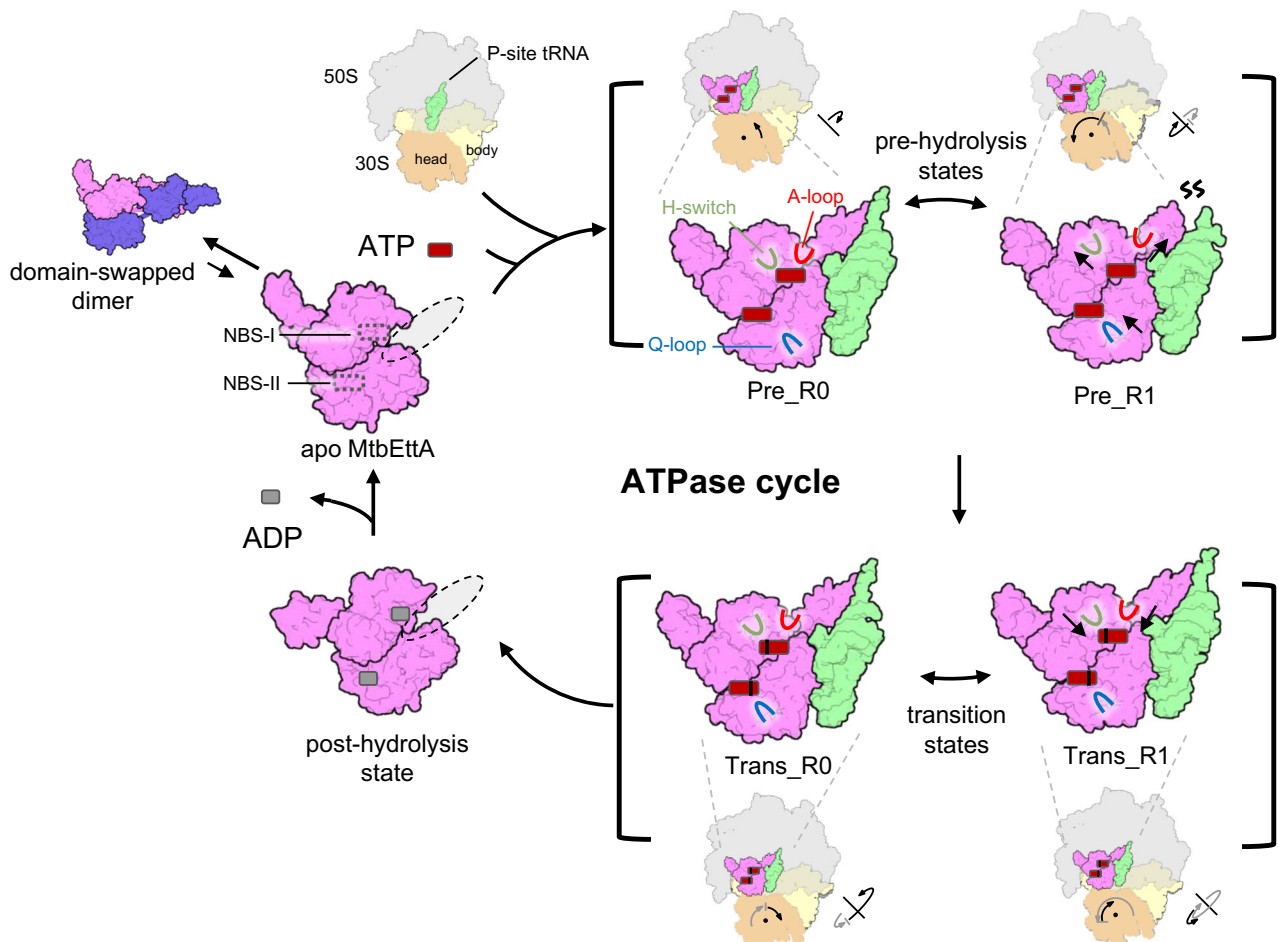

**Fig. 7 Schematic model for the functional cycle of MtbEttA in the course of ATP hydrolysis.** The monomer formation involves dissociation of the domain-swapped dimer (the favored configuration in solution) and rearrangement of the two NBDs to form a compact but open conformation. The monomeric MtbEttA transits to the closed conformation after binding to the E-site of the 70SIC in the presence of ATP molecules. At the pre-hydrolysis stage, the engagement of the Q-loop at NBS-II requires conformational change of the H-switch and A-loop in NBS-I, which is mediated by the bending of PtIM α1 along with the 30S rotation. In addition, the tip of PtIM and the CCA tail of tRNA become distorted. Meanwhile, remodeling of the 50S occurs as a result of the occupancy of MtbEttA at the E-site and the intermediate rotation of 30S. In the transition state, MtbEttA and the P-site tRNA maintain stable interactions due to a more restricted 30S movement. The head swiveling and body rotation of 30S are indicated with arrows for the movement in the previous (gray arrows) and current (black arrows) steps, respectively, in a sequence of nonrotated, Pre_R0, Pre_R1, Trans_R1, and Trans_R0 states. At the post-hydrolysis stage, MtbEttA adopts an open conformation and disassociates from the ribosome. ADP dissociation and ATP reloading are necessary for the next cycle.

transiently, which causes a conformational change of the MtbEttA, allowing it to exist in a more compact form (or closed state) to fit into the E site of the *Mtb* 70SIC. EttA-like proteins are very weak ATPases and cannot efficiently hydrolyze ATP in the absence of ribosomes[12,21]. However once absorbed into the E-site of the ribosome, the rotations of the 30S subunit facilitates the further engagement of the ATP-hydrolysis loops in the NBSs, fostering a more closed EttA to hydrolyze ATP.

Additionally, our results provide a structural explanation for the function of EttA-like proteins to throttle the translation depending on the ATP/ADP concentration ratio. At a higher ADP concentration, the ADP molecules may exchange with the bound ATP molecules in the MtbEttA. This may be possible, particularly in the pre-hydrolysis states where one of the two NBSs is not fully engaged onto the nucleotides. Future studies by single molecular experiments are necessary to elucidate the kinetics of the exchange between ATP and ADP at different ratios. While EcoEttA is not essential for *E. coli*[12], MtbEttA is essential for the growth and survival of *Mtb*. Besides its role in the translational regulation, we cannot rule out the possibility that

MtbEttA may be involved in another essential pathway. Nevertheless, due to the essentiality of MtbEttA, our high-resolution cryo-EM and crystal structures may provide frameworks for designing drugs to target this protein.

## Methods

**Protein expression and purification**. The full-length sequence of MtbEttA was cloned into a modified pET28(a) vector with an N-terminal His6-SUMO tag. We then overexpressed full-length MtbEttA in Rosetta™ 2(DE3) cells and purified it using Ni-NTA affinity chromatography. The SUMO tag was cleaved by SUMO protease and washed off from the Ni-NTA column, leaving MtbEttA bound with one extra serine at the N terminus. Further purification was performed by gel filtration chromatography through a Superdex 200 (16/60 GL) column (GE Healthcare) to separate the dimer from the monomer populations.

*Mtb* cells MC² 7000[36] were grown in 7H9 medium supplemented with 10% oleic albumin dextrose catalase (BD), 0.5% glycerol, 0.05% Tween-80, and 50 μg/ml pantothenic acid at 37 °C until an OD 600 of 1.0. The following procedures were performed at 4 °C. *Mtb* ribosomes were purified according to modified protocols[37]. After the cells were lysed in buffer (20 mM Tris-HCl [pH 7.5], 100 mM NH₄Cl, 10 mM MgCl₂, 0.5 mM EDTA, 6 mM 2-mercaptoethanol), it was clarified by centrifugation at 30,000 × *g* for 1 h. The supernatant was pelleted in sucrose cushion buffer (20 mM HEPES [pH 7.5], 1.1 M sucrose, 10 mM MgCl₂, 0.5 M KCl, and 0.5 mM EDTA) at 125,000 × *g* in a Beckman Type 45Ti rotor for 20 h. The

pellet was resuspended in the buffer containing 20 mM Tris-HCl (pH 7.5), 1.5 M $(NH_4)_2SO_4$, 0.4 M KCl, and 10 mM $MgCl_2$). The suspension was then applied to a hydrophobic interaction column (Toyopearl Butyl-650S) and eluted with a reverse ionic strength gradient from 1.5 M to 0 M $(NH_4)_2SO_4$ in the buffer containing 20 mM Tris-HCl (pH 7.5), 0.4 M KCl, and 10 mM $MgCl_2$. The eluted ribosome peak was changed to re-association buffer (5 mM HEPES-NaOH [pH 7.5], 10 mM $NH_4Cl$, 50 mM KCl, 10 mM $MgCl_2$, and 6 mM 2-mercaptoethanol) and concentrated before loading on top of a 10–40% linear sucrose gradient centrifuged in a Beckman SW28 rotor at $48,400 \times g$ for 19 h. The 70S fraction was concentrated to about A260 = 300 after removal of the sucrose.

**fMet-tRNA$^{fMet}$ and mRNA preparation**. *E. coli* tRNA$^{fMet}$ was expressed from an over-expression clone made in the pKK223-3 plasmid and was purified from *E. coli* cells upon IPTG-induction of the tac promoter[38]. The level of over-expression in the total tRNA was determined to be 40% by a label-free aminoacylation assay[39]. *E. coli* tRNA$^{fMet}$ was purified by hybridization to a biotin-conjugated complementary oligonucleotide bound to a streptavidin-coated resin. The tRNA was eluted from the resin by heat and re-annealed at 37 °C. Formylation of the tRNA (100 μM) was carried out during the aminoacylation reaction by including methionyl–tRNA formyltransferase (10 μM) with the methyl donor 10-formyltetrahydrofolate (0.85 mM; derived from folinic acid at neutral pH) in the presence of *E. coli* methionyl-tRNA synthetase (2.5 μM), Met (0.12 mM), and ATP (2 mM)[40].

**In-vitro translation assay**. The assay used to measure in-vitro ribosome activity relied on the production of Nanoluciferase in an *Mtb*-based cell-free system. *Mtb* S30 cell-free extract was prepared from *Mtb* MC$^2$7000, and equilibrated in S30 buffer (10 mM tris-acetate pH 8.2, 14 mM magnesium acetate, 60 mM potassium acetate, 1 mM DTT)[41]. 10 μL of S30 extract (containing 200 nM ribosome) was mixed with 5 μL 10X salt buffer (2 M potassium glutamate, 0.8 M ammonium acetate, and 0.16 M magnesium acetate), 1 mM each of the 20 amino acids and 33 mM phosphoenolpyruvate to a final volume of 41 μL. We then added the erythromycin of various concentrations (2 μL) to the 41 μL mixture, and after the incubation for 10 min at RT, the reaction was supplemented with MtbEttA, Nanoluciferase mRNA (200 ng), and 5 μL master mix (286 mM HEPES-KOH, pH7.5, 6 mM ATP, 4.3 mM GTP, 333 μM folinic acid, 853 μg/mL tRNA). The final volume was 50 μL in each well of the 384-well plate. The reaction proceeded for 40 min at 37° at which time the reaction was terminated by the addition of 80 μM Chloramphenicol. The luminescent signal was detected by the addition of 20 μL of the nano-luciferase substrate Furimazine (Promega). Nanoluciferase mRNA was prepared from an in-vitro transcription assay[42].

**Cryo-EM sample preparation and data collection**. The *Mtb* 70SIC in complex with MtbEttA at the pre-hydrolysis state was prepared with 0.2 μM 70S, 0.2 μM modified Z4C mRNA (AGAAAGGAGGUAAAACAUGUUCAAAA)[42], 0.8 μM fMet-tRNA$^{fMet}$, 33 μM MtbEttA, and 2 mM ADPNP. Firstly, the 70SIC containing fMet-tRNA$^{fMet}$ and the mRNA was formed in buffer A (50 mM HEPES pH 7.5, 50 mM KCl, 10 mM NH4Cl, 5 mM $MgCl_2$, 6 mM β-mercaptoethanol) at 37 °C for 30 min, and then supplied with fMet-tRNA$^{fMet}$ for another 30 min at 37 °C. In parallel, MtbEttA was mixed with ADPNP at 37 °C for 1 h. The final complex was formed by combining the two mixtures together and bringing the $Mg^{2+}$ concentration to 10 mM, incubated at 37 °C for 1 h, and then kept on ice until use. The *Mtb* 70SIC with MtbEttA at transition state was prepared in the similar fashion, except that 4 mM ADP and 4 mM $Na_3VO_4$ were used instead of ADPNP. Cryo-EM specimens were then prepared by applying 3 μL of a freshly reconstituted complex to a glow-discharged Quantifoil 2/1 200-mesh Holey Carbon Grid coated with 2 nm continuous carbon, and vitrified using a Vitrobot Mark III (Thermo Fisher Scientific) at 22 °C with 100% relative humidity.

Cryo-EM images of the 70SIC-MtbEttA-ADPNP complex were recorded under a Titan Krios microscope (Thermo Fisher Scientific) operated at 300 kV. Data were collected using EPU on a K2 Summit direct detection camera (Gatan) in the electron counting mode with a pixel size of 1.06 Å. Beam shift was enabled to encompass 5 exposures per hole. The beam intensity was adjusted to a dose rate of 5 e$^-$ per pixel per second on the camera. A 30-frame movie stack was recorded for each exposure with 0.2 s per frame for a total exposure time of 6 s. A post-column energy filter was used with a slit width of 20 eV.

Similarly, 70SIC-MtbEttA-ADP-VO$_4$ images were recorded under another Titan Krios microscope operated at 300 kV. Data were collected using EPU on a K2 Summit direct detection camera (Gatan) in the electron counting mode with a pixel size of 1.063 Å. Beam shift was enabled to encompass 4 exposures per hole. The beam intensity was adjusted to a dose rate of 6.5 e$^-$ per pixel per second on the camera. A 30-frame movie stack was recorded for each exposure with 0.2 s per frame for a total exposure time of 6 s. A post-column energy filter was used with a slit width of 20 eV.

**Image preprocessing**. Drift correction of collected movie stacks was done using MotionCor2[43]. The defocus value of each aligned micrograph was determined using Gctf[44]. Micrographs with visible contamination and poor power spectrum were discarded. Automatic particle picking was done by gautomatch (Zhang

software, MRC Laboratory of Molecular Biology). 3D reconstruction was done by following the pipeline of Relion3[45].

Data processing for the 70SIC-MtbEttA-ADPNP sample was achieved as follows (Supplementary Fig. 3): In total, 1,175,176 particles were selected from 8,949 micrographs and binned by 8 before subjecting to 2D classifications in Relion to remove bad particles. After 2D classifications, 1,016,405 particles were selected and binned by 4. The initial 3D refinement step was performed with all clean particles to get a consensus map. 3D classification using the consensus map as initial model and with the "skip_align" option was used to classify different subpopulations in the dataset. Two out of eight classes were found with the density for MtbEttA. A total of 196,100 particles were selected after an additional cleaning step, using focused 3D classification without alignment around MtbEttA and the P-site tRNA. Further 3D classification of these particles with the "skip_align" option and 30S mask yielded 3 subpopulations, including the Pre_R0 state with 126,715 particles and the Pre_R1 state with 34,158 particles. We continued to refine the Pre_R0 and Pre_R1 states with unbinned data, to the resolutions of 2.97 and 3.23 Å, respectively. Different subpopulations including 70S-P/PtRNA, 70S-P/EtRNA, 70S-P/PtRNA-E/EtRNA and 50S were also refined to resolutions of 2.76, 2.8, 2.71, and 3.0 Å, respectively.

Similar to the procedure above, 70SIC-MtbEttA-ADP-VO$_4$ data were processed accordingly (Supplementary Fig. 5): After 2D classifications, 742,504 particles from 8,666 micrographs were selected, and the 3D classification without alignment was performed using a consensus map obtained from an initial 3D refinement step. Two out of six classes were found with clear MtbEttA density. A total of 125,420 particles were selected after an additional cleaning step, using focused 3D classification without alignment around MtbEttA or the P-site tRNA. Further 3D classification of these particles with the "skip_align" option and 30S mask yielded 3 subpopulations, including the Trans_R0 state with 86,692 particles and the Trans_R1 state with 32,731 particles. We continued to refine the Trans_R0 and Trans_R1 states with unbinned data, to the resolutions of 2.79 and 3.1 Å, respectively.

**Resolution estimation and post processing**. The overall resolution of all these reconstructed maps was assessed using the gold-standard criterion of Fourier shell correlation[46], with a cutoff at 0.143, between two half-maps from two independent half-sets of data. Local resolutions were estimated using Resmap[47]. Post processing was done using LocalDeblur[48].

**Crystallization, data collection, and structure determination of MtbEttA-ADP**. Purified full-length MtbEttA was concentrated to ~46 mg/ml. Prior to crystallization, the final concentration of 4 mM ADP and 4 mM $MgCl_2$ were mixed with the protein for 1 h at room temperature. Crystals were observed at multiple conditions at 16 °C with initial screening against ~600 conditions, using the sitting-drop vapor diffusion set by a Mosquito Crystal liquid handler (TTP Labtech Inc). The diffraction quality of crystals was checked at the in-house X-ray source. Further optimization was performed with the hanging drop vapor diffusion method, yielding the best crystal at the condition of 0.1 M MES pH 6.7, 0.2 M $MgCl_2$, 10% v/v PEG 4000.

Diffraction data were collected at the Advance Photon Source, Argonne National Laboratory in Chicago. Crystals diffracted to ~2.5 Å, with deterioration in the later frames. Diffraction data were scaled and truncated to 2.9 Å using imosflm[49]. Molecular replacement was performed with AutoMR[50] in the PHENIX package[51], using two NBDs models of MtbEttA predicted from SWISS-MODEL[52] based on the crystal structure of EcoEttA (PDB 4FIN [https://www.rcsb.org/structure/4fin]). The model was iteratively refined and manually built with PHENIX and ISOLDE[53], respectively.

**Modeling and visualization**. To build the atomic model for 70SIC-MtbEttA-ADPNP and 70SIC-MtbEttA-ADP-VO4 complexes, we first fit our previous *Mtb* 70S ribosome structure (PDB 5V93 [https://www.rcsb.org/structure/5v93]) and MtbEttA monomer structure obtained from SWISS-MODEL into the high-resolution cryo-EM map as rigid body using University of California San Francisco (UCSF) Chimera[54]. Model refinement was performed by real-space refinement in PHENIX. RNA geometry optimization was done by ERRASER[55]. Manual model building was done with COOT[56] to inspect and improve local fitting. The iterative process of refinement and the manual building was conducted to achieve the best model. The same model building procedure was also done for 70S-P/PtRNA, 70S-P/EtRNA, and 70S-P/PtRNA-E/EtRNA complexes. All of the figures and movies were made using UCSF Chimera and ChimeraX[57].

**Reporting summary**. Further information on research design is available in the Nature Research Reporting Summary linked to this article.

## Data availability
The data that support this study are available from the corresponding author upon reasonable request. The cryo-EM maps and models of *Mtb* 70SIC with MtbEttA at Pre_R0, Pre_R1, Trans_R0, Trans_R1 states and *Mtb* 70S with P/P tRNA, P/E tRNA, P/P, and E/E tRNAs are deposited in the EMData Bank with accession codes EMD-23961,

EMD-23962, EMD-23969, EMD-23972, EMD-23974, EMD-23975, and EMD-23976, and in the Protein Data Bank with accession codes 7MSC, 7MSH, 7MSM, 7MSZ, 7MT2, 7MT3 and 7MT7, respectively. The cryo-EM map of *Mtb* 50S is deposited in the EMData Bank with accession ID EMD-23981. The model of ADP-bound MtbEttA is deposited in the Protein Data Bank with accession ID 7MU0. The previous structures of the *Mtb* 70S ribosome and EcoEttA, which facilitated our modeling, can be accessed from the Protein Data Bank with accession 5V93 [] and 4FIN [], respectively. Source data are provided with this paper.

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

## Acknowledgements

We thank the Microscopy and Imaging Center at Texas A&M University for providing instrumentation for cryo-EM sample screening and the Texas A&M High-Performance Research Computing Center for providing the computational resources for data processing. J.Z. is supported by start-up funding from the Department of Biochemistry and Biophysics at Texas A&M University and the Center for Phage Technology, jointly sponsored by Texas AgriLife and Texas A&M University. J.Z. acknowledges the X-grant from Texas A&M, the Welch Foundation grants A-1863, the NSF grant MCB-1902392, and the NIH grants R21AI137696, R21AI156846, and U24GM116787. J.Z. and J.C.S. are both supported by the NIH grant P01AI095208.

## Author contributions

Z.C. and J.Z. designed research; Z.C., X.L., and J.S. performed research; Z.C., X.L., J.S., and J.Z. analyzed data; H.G. and Y.H. provided reagents; and all authors wrote the paper.

## Competing interests

The authors declare no competing interests.
