## [Peer Review File · Nature Communications]

Interplay between an ATP-binding cassette F protein and the ribosome from *Mycobacterium tuberculosis*Reviewers' Comments:

Reviewer #1:

Remarks to the Author:

The manuscript by Cui et al, entitled "Interplay between an ATP-binding cassette F protein and the ribosome from *Mycobacterium tuberculosis*", describes structural features of a homolog of ABC-F protein EttA in ribosome bound and ribosome free forms. The results expanded more features regarding EttA family proteins in association with ribosome, could offer certain merits to the field. Given two papers regarding EttA (both functional and structural data) published in NSMB, here there is not much compelling finding.

As for the target protein (Rv2477c) from *Mycobacterium*, the author claimed that it is essential in the strain, this raised an issue: EttA proteins or AMR ABC-F proteins usually are not essential under normal condition, why is it essential in this case? But following the text "essential for cell growth based on a study by saturating transposon mutagenesis24 (<https://mycobrowser.epfl.ch/genes/Rv2477c>)", it seems that this is not correct description for this protein. I encourage the author to confirm this.

1. Suppose sequence identity and similarity is quite high between MtbEttA and EcoEttA, the author should include a sequence comparison. Subsequently, it would be interest to compare the conservation for those key residues, as well as to compare their functions (EcoEttA was biochemically characterized, such as a role in initial step of protein synthesis).
2. The authors identified some important interactions for MtbEttA with the Mtb ribosome, such as R280 in MtbEttA etc, any functional and mutational data on these proposed important residues?
3. The author indicated monomer and homodimer formation subject to salt concentration (like 500 mM salt). However, the author did not test whether dimer formation could be affected by protein concentration (in a concentration manner?). Furthermore, the author also tested the influence of nucleotides in Fig.6a. Collectively, what is the physiological relevance?
4. In Fig. 6, the author compared structure of ADP bound MtbEttA with others. However, how about the comparison of MtbEttA with EcoEttA?
5. The authors claimed that the resolved structures represent Mtb ribosome with ADPNP and ADP-VO₄, because they form these two complexes and EM map demonstrated the nucleotides (Extended Data Fig. 11), respectively. However, the authors should use a map with mesh, rather than that used in figure which is not so sharp and is easy to be confused with atom surface. It is quite uncertain whether ADP-VO₄ in the MtbEttA with ribosome can be trapped through a simple mixing step as the authors described. This is very critical, it should be validated.
6. Again, the authors should demonstrate the map as well for ADP in crystal structure of the ADP bound state.
7. The crystallographic table should include Ramachandran plot.

Reviewer #2:

Remarks to the Author:

In their manuscript, Cui and co-authors aim to describe a structural mechanism of *M. tuberculosis* ABC-F protein Etta. This protein was previously found to affect translation efficiency, but understanding of the structural mechanisms of Etta and the role of ATPase activity required high-resolution structures. The authors present a panel of high-resolution ~3 Å structures of Etta bound to a 70S ribosome in the presence of non-hydrolyzable ATP analog or a transition state analog (ADP + vanadate). The authors have also crystallized *M.tub* Etta with ADP. Together, these structures provide an insight into the mechanism of Etta function in stabilizing aa-tRNA in the P site to promote translation. The work is well presented, and the findings overall support the mechanism that the authors propose. There are minor suggestions that I recommend the authors to address in the revised manuscript:

1. The chemically correct form of the vanadate ion is VO₄³⁻ (the charge is specified in superscript),

this should be corrected throughout the manuscript.

2. The authors argue that EttA is unlikely to be an antibiotic-resistance gene (consistent with the 2014 NSMB paper on Etta). But recent structures of Lsa and Vga also seem to have shorter ARD loops than VmlR and MsrE (which the authors do compare EttA with), yet they confer antibiotic resistance. It would help to add structural comparison of ribosome-bound MtbEtta with the antibiotic-resistance-inducing ATPases Lsa and Vga (ref: <https://www.nature.com/articles/s41467-021-23753-1>). Perhaps this would suggest that an antibiotic-resistance function of Etta should not be ruled out.

3. The P-tRNA stabilization role of Etta makes sense, based on the presented cryo-EM structures. This is similar to the functions of bacterial EF-P (ref: [https://www.cell.com/molecular-cell/pdf/S1097-2765\(17\)30789-X.pdf](https://www.cell.com/molecular-cell/pdf/S1097-2765(17)30789-X.pdf)) and eukaryotic eIF5A (<https://www.ncbi.nlm.nih.gov/pmc/articles/PMC4770232/>). This similarity is worth discussing in the manuscript.

4. What is the mechanistic implication and significance of MtbEttA structure without nucleotide? ("The trial to obtain a crystal of the nucleotide-free MtbEttA was unsuccessful. We then predicted 427 the structure of the nucleotide-free MtbEttA using homology modelling based on the crystal 428 structure of EcoEttA in the ATP-free state."). This part describes a predicted structure, whose relevance is not clear, so it is not necessary.

5. The meaning of the term "throttle the translation" is not clear. Please describe the function in molecular terms.

6. In Discussion, the authors propose a role of Etta in A-site tRNA accommodation by displacing H89 proposed by a theoretical study (ref. 29). Cryo-EM of accommodating tRNA showed that A-tRNA accommodation is facilitated by 30S rotation (ref: <https://pubmed.ncbi.nlm.nih.gov/32612237/>). The authors therefore should mention that Etta sampling different rotations of 30S is also consistent with assisting in tRNA accommodation observed experimentally.

Reviewer #1 (Remarks to the Author):

The manuscript by Cui et al, entitled “Interplay between an ATP-binding cassette F protein and the ribosome from *Mycobacterium tuberculosis*”, describes structural features of a homolog of ABC-F protein EttA in ribosome bound and ribosome free forms. The results expanded more features regarding EttA family proteins in association with ribosome, could offer certain merits to the field. Given two papers regarding EttA (both functional and structural data) published in NSMB, here there is not much compelling finding.

As for the target protein (Rv2477c) from *Mycobacterium*, the author claimed that it is essential in the strain, this raised an issue: EttA proteins or AMR ABC-F proteins usually are not essential under normal condition, why is it essential in this case? But following the text “essential for cell growth based on a study by saturating transposon mutagenesis24 (<https://mycobrowser.epfl.ch/genes/Rv2477c>)”, it seems that this is not correct description for this protein. I encourage the author to confirm this.

We thank the reviewer for pointing this out. We confirm that Rv2477c is MtbEttA. On the mycobrowser website, Rv2477c is annotated as “Probable macrolide-transport ATP-binding protein ABC transporter”. However so far, such a function of a transporter has not been observed. Moreover, although Rv2477c belongs to the ABC superfamily, it does not contain a transmembrane domain to function as an ABC transporter. Therefore, it is more likely to be a ribosomal factor as shown in our manuscript as well as its high homology to the EcoEttA.

1. Suppose sequence identity and similarity is quite high between MtbEttA and EcoEttA, the author should include a sequence comparison. Subsequently, it would be interest to compare the conservation for those key residues, as well as to compare their functions (EcoEttA was biochemically characterized, such as a role in initial step of protein synthesis).

We have made a new Extended Data Fig. 8 for the sequence comparison of nine EttA-like proteins from different species (including EcoEttA). Based on this sequence comparison, all the key residues in the Walker A, Walker B, H-switch, Q-loop, A-loop and ABC signature motif are mostly conserved, as well as the positively-charged residues at the tip of the PtIM, which strongly suggest a functional conservation of these residues in ATP binding/hydrolysis and the interaction with the RNA backbone. Notably, the aromatic residues in the A-loop of the NBS1 for EcoEttA are missing, replaced by two prolines, which may replace large aromatic residues to form ring-stacking with the adenine ring of ATP.

2. The authors identified some important interactions for MtbEttA with the Mtb ribosome, such as R280 in MtbEttA etc, any functional and mutational data on these proposed important residues?

Unfortunately, there is no previously published mutagenesis data on these residues. We tried to perform the mutagenesis of these residues, however, the purification of all these mutant proteins were not successful, including the R280 mutations which fell into aggregations. This is

very likely due to the key locations of these residues that maintain the stability of the protein. Nevertheless, as shown in our Extended Data Fig. 8, the R280 of PtlM is highly conserved among different EttA-like proteins, which strongly suggest its functional role in interacting with the CCA tail of the P-site tRNA. We have added the following statement in the result section where we describe the **Interactions between MtbEttA and the *Mtb* ribosome**:

“Interestingly, this arginine residue is conserved among different EttA-like proteins (Extended Data Fig. 8), suggesting a functional role in its interaction with the neighboring RNA backbones.”

3. The author indicated monomer and homodimer formation subject to salt concentration (like 500 mM salt). However, the author did not test whether dimer formation could be affected by protein concentration (in a concentration manner?). Furthermore, the author also tested the influence of nucleotides in Fig.6a. Collectively, what is the physiological relevance?

We thank the reviewer for the comment. We performed the gel-filtration chromatography of MtbEttA at different protein concentrations (1mg/ml and 0.2mg/ml) to see the relative population of dimer and monomers (the new Extended Data Fig.16). It appears that MtbEttA exists dominantly as dimers regardless of the difference in the tested protein concentrations.

It also appears that such a dominance of dimers is not affected by the nucleotide. Under physiological concentration of the salt (~250 mM NaCl) in an *Mtb* cell, the MtbEttA would exist mostly as dimers. Therefore, it is also highly likely that *in vivo*, they exist as dimers.

To make a connection to the physiology of the *Mtb*, we propose that the large population of the inactive dimers may serve as a reservoir of MtbEttA, which will dissociate into monomers to interact with the *Mtb* ribosomes. Such a hypothesis for the dimer/monomer switch needs to be tested. It is possible that the presence of the *Mtb* ribosome binding the monomeric MtbEttA drives the dimer/monomer equilibrium towards the monomers. Admittedly, we cannot rule out that, *in vivo*, there exists a potential factor to regulate the oligomeric states of MtbEttA, which is yet to be discovered.”

We have added the above hypothesis in the first paragraph of the Discussion section.

4. In Fig. 6, the author compared structure of ADP bound MtbEttA with others. However, how about the comparison of MtbEttA with EcoEttA?

Since the high-resolution structure of EcoEttA was crystallized in a nucleotide-free state, we have now made a new Extended Data Fig. 15. to compare the crystal structures of ADP-bound MtbEttA and nucleotide-free EcoEttA. In the crystal, the asymmetric unit of MtbEttA-ADP consists of a domain-swapped dimer, similar to that in the crystal structure of nucleotide-free EcoEttA (Extended Data Fig. 15a). Apparently, the conformations of both NBSs in the nucleotide-free EcoEttA are not compatible for stable nucleotide binding, as Pro16 and Tyr333 in the A-loop of NBS1 and NBS2, respectively, are either too far from or collide with the adenine ring of a potential nucleotide (Extended Data Fig. 15b,c). On the contrary, the corresponding residues, His15 and Tyr331, in the ADP-bound MtbEttA stack with the adenine ring of ADP, with

the density of the ADP and magnesium ions clearly resolved in the crystal structure (Extended Data Fig. 15d,e).

We have added the above description for the structural comparison in the first paragraph of the section where we show “The crystal structure of ribosome-free MtbEttA at the post-hydrolysis (ADP) state”.

5. The authors claimed that the resolved structures represent Mtb ribosome with ADPNP and ADP-VO₄, because they form these two complexes and EM map demonstrated the nucleotides (Extended Data Fig. 11), respectively. However, the authors should use a map with mesh, rather than that used in figure which is not so sharp and is easy to be confused with atom surface. It is quite uncertain whether ADP-VO₄ in the MtbEttA with ribosome can be trapped through a simple mixing step as the authors described. This is very critical, it should be validated.

We thank the reviewer for the suggestions and have revised the Extended Data Fig. 11 (which is now the new Extended Data Fig. 12) with mesh for nucleotide densities. The ADP and VO₄³⁻ densities are clearly visible, confirming the ADP-VO₄ complex in our MtbEttA.

6. Again, the authors should demonstrate the map as well for ADP in crystal structure of the ADP bound state.

We have added two new panels in Extended Data Fig.15 (Panels d and e) to show the electron density for ADP in the crystal structure of the ADP-bound MtbEttA.

7. The crystallographic table should include Ramachandran plot.

We have added the Ramachandran plot right below the crystallographic table.

Reviewer #2 (Remarks to the Author):

In their manuscript, Cui and co-authors aim to describe a structural mechanism of *M. tuberculosis* ABC-F protein Etta. This protein was previously found to affect translation efficiency, but understanding of the structural mechanisms of Etta and the role of ATPase activity required high-resolution structures. The authors present a panel of high-resolution ~3 Å structures of Etta bound to a 70S ribosome in the presence of non-hydrolyzable ATP analog or a transition state analog (ADP + vanadate). The authors have also crystallized *M.tub* Etta with ADP. Together, these structures provide an insight into the mechanism of Etta function in stabilizing aa-tRNA in the P site to promote translation. The work is well presented, and the findings overall support the mechanism that the authors propose. There are minor suggestions that I recommend the authors to address in the revised manuscript:

1. The chemically correct form of the vanadate ion is VO_4^{3-} (the charge is specified in superscript), this should be corrected throughout the manuscript.

We appreciate the reviewer for pointing this out. Yes, the vanadate ion should be annotated as VO_4^{3-} when referring to the vanadate ion alone. However, in our text and figures, we refer to the ADP- VO_4 , which are commonly annotated as it is.

2. The authors argue that EttA is unlikely to be an antibiotic-resistance gene (consistent with the 2014 NSMB paper on Etta). But recent structures of Lsa and Vga also seem to have shorter ARD loops than VmlR and MsrE (which the authors do compare EttA with), yet they confer antibiotic resistance. It would help to add structural comparison of ribosome-bound MtbEtta with the antibiotic-resistance-inducing ATPases Lsa and Vga (ref: <https://www.nature.com/articles/s41467-021-23753-1>). Perhaps this would suggest that an antibiotic-resistance function of Etta should not be ruled out.

We added the structural comparison of PtiM in MtbEttA with ARD loops in LsaA and VgaL (see Extended Data Fig.6 f,g). Even though the ARD loops of LsaA and VgaL are shorter than the ones in VmlR and MsrE, they are still longer than the PtiM in MtbEttA. Hence, this reinforces our claim that MtbEttA is unlikely to have an antibiotic-resistance function in terms of the PtiM length.

3. The P-tRNA stabilization role of Etta makes sense, based on the presented cryo-EM structures. This is similar to the functions of bacterial EF-P (ref: [https://www.cell.com/molecular-cell/pdf/S1097-2765\(17\)30789-X.pdf](https://www.cell.com/molecular-cell/pdf/S1097-2765(17)30789-X.pdf)) and eukaryotic eIF5A (<https://www.ncbi.nlm.nih.gov/pmc/articles/PMC4770232/>). This similarity is worth discussing in the manuscript.

We thank the reviewer for this comment and have added a sentence in the 3rd paragraph of the discussion section of the main text to refer to EF-P and eIF5A, and cited the two papers mentioned above.

4. What is the mechanistic implication and significance of MtbEttA structure without nucleotide? (“The trial to obtain a crystal of the nucleotide-free MtbEttA was unsuccessful. We then predicted 427 the structure of the nucleotide-free MtbEttA using homology modelling based on the crystal 428 structure of EcoEttA in the ATP-free state.”). This part describes a predicted structure, whose relevance is not clear,

We thank the reviewer for the suggestions and removed this paragraph and the corresponding supplement figure.

5. The meaning of the term “throttle the translation” is not clear. Please describe the function in molecular terms.

The word “throttle” is proposed by the previous literature which characterized the function of E. coli EttA (Energy-dependent translational throttle protein), which regulates the ribosome’s entry into the translation elongation cycle, depending on the ATP/ADP ratio. We have further stated it in the first paragraph of the Main section.

6. In Discussion, the authors propose a role of Etta in A-site tRNA accommodation by displacing H89 proposed by a theoretical study (ref. 29). Cryo-EM of accommodating tRNA showed that A-tRNA accommodation is facilitated by 30S rotation (ref: <https://pubmed.ncbi.nlm.nih.gov/32612237/>). The authors therefore should mention that Etta sampling different rotations of 30S is also consistent with assisting in tRNA accommodation observed experimentally.

We thank the reviewer for the suggestion. We have added a sentence in which this cryo-EM study of experimentally observed tRNA accommodation into A-site supports our result. We have also cited this paper in our revised manuscript.

Reviewers' Comments:

Reviewer #1:

Remarks to the Author:

The authors have addressed all my comments

Reviewer #2:

Remarks to the Author:

The manuscript has substantially improved, and I am satisfied with how the authors addressed reviewers' comments. I recommend to publish this well written manuscript.

Reviewer #1 (Remarks to the Author):

The manuscript by Cui et al, entitled “Interplay between an ATP-binding cassette F protein and the ribosome from *Mycobacterium tuberculosis*”, describes structural features of a homolog of ABC-F protein EttA in ribosome bound and ribosome free forms. The results expanded more features regarding EttA family proteins in association with ribosome, could offer certain merits to the field. Given two papers regarding EttA (both functional and structural data) published in NSMB, here there is not much compelling finding.

As for the target protein (Rv2477c) from *Mycobacterium*, the author claimed that it is essential in the strain, this raised an issue: EttA proteins or AMR ABC-F proteins usually are not essential under normal condition, why is it essential in this case? But following the text “essential for cell growth based on a study by saturating transposon mutagenesis24 (<https://mycobrowser.epfl.ch/genes/Rv2477c>)”, it seems that this is not correct description for this protein. I encourage the author to confirm this.

We thank the reviewer for pointing this out. We confirm that Rv2477c is MtbEttA. On the mycobrowser website, Rv2477c is annotated as “Probable macrolide-transport ATP-binding protein ABC transporter”. However so far, such a function of a transporter has not been observed. Moreover, although Rv2477c belongs to the ABC superfamily, it does not contain a transmembrane domain to function as an ABC transporter. Therefore, it is more likely to be a ribosomal factor as shown in our manuscript as well as its high homology to the EcoEttA.

1. Suppose sequence identity and similarity is quite high between MtbEttA and EcoEttA, the author should include a sequence comparison. Subsequently, it would be interest to compare the conservation for those key residues, as well as to compare their functions (EcoEttA was biochemically characterized, such as a role in initial step of protein synthesis).

We have made a new Extended Data Fig. 8 for the sequence comparison of nine EttA-like proteins from different species (including EcoEttA). Based on this sequence comparison, all the key residues in the Walker A, Walker B, H-switch, Q-loop, A-loop and ABC signature motif are mostly conserved, as well as the positively-charged residues at the tip of the PtIM, which strongly suggest a functional conservation of these residues in ATP binding/hydrolysis and the interaction with the RNA backbone. Notably, the aromatic residues in the A-loop of the NBS1 for EcoEttA are missing, replaced by two prolines, which may replace large aromatic residues to form ring-stacking with the adenine ring of ATP.

2. The authors identified some important interactions for MtbEttA with the Mtb ribosome, such as R280 in MtbEttA etc, any functional and mutational data on these proposed important residues?

Unfortunately, there is no previously published mutagenesis data on these residues. We tried to perform the mutagenesis of these residues, however, the purification of all these mutant proteins were not successful, including the R280 mutations which fell into aggregations. This is

very likely due to the key locations of these residues that maintain the stability of the protein. Nevertheless, as shown in our Extended Data Fig. 8, the R280 of PtlM is highly conserved among different EttA-like proteins, which strongly suggest its functional role in interacting with the CCA tail of the P-site tRNA. We have added the following statement in the result section where we describe the **Interactions between MtbEttA and the *Mtb* ribosome**:

“Interestingly, this arginine residue is conserved among different EttA-like proteins (Extended Data Fig. 8), suggesting a functional role in its interaction with the neighboring RNA backbones.”

3. The author indicated monomer and homodimer formation subject to salt concentration (like 500 mM salt). However, the author did not test whether dimer formation could be affected by protein concentration (in a concentration manner?). Furthermore, the author also tested the influence of nucleotides in Fig.6a. Collectively, what is the physiological relevance?

We thank the reviewer for the comment. We performed the gel-filtration chromatography of MtbEttA at different protein concentrations (1mg/ml and 0.2mg/ml) to see the relative population of dimer and monomers (the new Extended Data Fig.16). It appears that MtbEttA exists dominantly as dimers regardless of the difference in the tested protein concentrations.

It also appears that such a dominance of dimers is not affected by the nucleotide. Under physiological concentration of the salt (~250 mM NaCl) in an *Mtb* cell, the MtbEttA would exist mostly as dimers. Therefore, it is also highly likely that *in vivo*, they exist as dimers.

To make a connection to the physiology of the *Mtb*, we propose that the large population of the inactive dimers may serve as a reservoir of MtbEttA, which will dissociate into monomers to interact with the *Mtb* ribosomes. Such a hypothesis for the dimer/monomer switch needs to be tested. It is possible that the presence of the *Mtb* ribosome binding the monomeric MtbEttA drives the dimer/monomer equilibrium towards the monomers. Admittedly, we cannot rule out that, *in vivo*, there exists a potential factor to regulate the oligomeric states of MtbEttA, which is yet to be discovered.”

We have added the above hypothesis in the first paragraph of the Discussion section.

4. In Fig. 6, the author compared structure of ADP bound MtbEttA with others. However, how about the comparison of MtbEttA with EcoEttA?

Since the high-resolution structure of EcoEttA was crystallized in a nucleotide-free state, we have now made a new Extended Data Fig. 15. to compare the crystal structures of ADP-bound MtbEttA and nucleotide-free EcoEttA. In the crystal, the asymmetric unit of MtbEttA-ADP consists of a domain-swapped dimer, similar to that in the crystal structure of nucleotide-free EcoEttA (Extended Data Fig. 15a). Apparently, the conformations of both NBSs in the nucleotide-free EcoEttA are not compatible for stable nucleotide binding, as Pro16 and Tyr333 in the A-loop of NBS1 and NBS2, respectively, are either too far from or collide with the adenine ring of a potential nucleotide (Extended Data Fig. 15b,c). On the contrary, the corresponding residues, His15 and Tyr331, in the ADP-bound MtbEttA stack with the adenine ring of ADP, with

the density of the ADP and magnesium ions clearly resolved in the crystal structure (Extended Data Fig. 15d,e).

We have added the above description for the structural comparison in the first paragraph of the section where we show “The crystal structure of ribosome-free MtbEttA at the post-hydrolysis (ADP) state”.

5. The authors claimed that the resolved structures represent Mtb ribosome with ADPNP and ADP-VO₄, because they form these two complexes and EM map demonstrated the nucleotides (Extended Data Fig. 11), respectively. However, the authors should use a map with mesh, rather than that used in figure which is not so sharp and is easy to be confused with atom surface. It is quite uncertain whether ADP-VO₄ in the MtbEttA with ribosome can be trapped through a simple mixing step as the authors described. This is very critical, it should be validated.

We thank the reviewer for the suggestions and have revised the Extended Data Fig. 11 (which is now the new Extended Data Fig. 12) with mesh for nucleotide densities. The ADP and VO₄³⁻ densities are clearly visible, confirming the ADP-VO₄ complex in our MtbEttA.

6. Again, the authors should demonstrate the map as well for ADP in crystal structure of the ADP bound state.

We have added two new panels in Extended Data Fig.15 (Panels d and e) to show the electron density for ADP in the crystal structure of the ADP-bound MtbEttA.

7. The crystallographic table should include Ramachandran plot.

We have added the Ramachandran plot right below the crystallographic table.

Reviewer #2 (Remarks to the Author):

In their manuscript, Cui and co-authors aim to describe a structural mechanism of *M. tuberculosis* ABC-F protein Etta. This protein was previously found to affect translation efficiency, but understanding of the structural mechanisms of Etta and the role of ATPase activity required high-resolution structures. The authors present a panel of high-resolution ~3 Å structures of Etta bound to a 70S ribosome in the presence of non-hydrolyzable ATP analog or a transition state analog (ADP + vanadate). The authors have also crystallized *M.tub* Etta with ADP. Together, these structures provide an insight into the mechanism of Etta function in stabilizing aa-tRNA in the P site to promote translation. The work is well presented, and the findings overall support the mechanism that the authors propose. There are minor suggestions that I recommend the authors to address in the revised manuscript:

1. The chemically correct form of the vanadate ion is VO_4^{3-} (the charge is specified in superscript), this should be corrected throughout the manuscript.

We appreciate the reviewer for pointing this out. Yes, the vanadate ion should be annotated as VO_4^{3-} when referring to the vanadate ion alone. However, in our text and figures, we refer to the ADP- VO_4 , which are commonly annotated as it is.

2. The authors argue that EttA is unlikely to be an antibiotic-resistance gene (consistent with the 2014 NSMB paper on Etta). But recent structures of Lsa and Vga also seem to have shorter ARD loops than VmlR and MsrE (which the authors do compare EttA with), yet they confer antibiotic resistance. It would help to add structural comparison of ribosome-bound MtbEtta with the antibiotic-resistance-inducing ATPases Lsa and Vga (ref: <https://www.nature.com/articles/s41467-021-23753-1>). Perhaps this would suggest that an antibiotic-resistance function of Etta should not be ruled out.

We added the structural comparison of PtIM in MtbEttA with ARD loops in LsaA and VgaL (see Extended Data Fig.6 f,g). Even though the ARD loops of LsaA and VgaL are shorter than the ones in VmlR and MsrE, they are still longer than the PtIM in MtbEttA. Hence, this reinforces our claim that MtbEttA is unlikely to have an antibiotic-resistance function in terms of the PtIM length.

3. The P-tRNA stabilization role of Etta makes sense, based on the presented cryo-EM structures. This is similar to the functions of bacterial EF-P (ref: [https://www.cell.com/molecular-cell/pdf/S1097-2765\(17\)30789-X.pdf](https://www.cell.com/molecular-cell/pdf/S1097-2765(17)30789-X.pdf)) and eukaryotic eIF5A (<https://www.ncbi.nlm.nih.gov/pmc/articles/PMC4770232/>). This similarity is worth discussing in the manuscript.

We thank the reviewer for this comment and have added a sentence in the 3rd paragraph of the discussion section of the main text to refer to EF-P and eIF5A, and cited the two papers mentioned above.

4. What is the mechanistic implication and significance of MtbEttA structure without nucleotide? (“The trial to obtain a crystal of the nucleotide-free MtbEttA was unsuccessful. We then predicted 427 the structure of the nucleotide-free MtbEttA using homology modelling based on the crystal 428 structure of EcoEttA in the ATP-free state.”). This part describes a predicted structure, whose relevance is not clear,

We thank the reviewer for the suggestions and removed this paragraph and the corresponding supplement figure.

5. The meaning of the term “throttle the translation” is not clear. Please describe the function in molecular terms.

The word “throttle” is proposed by the previous literature which characterized the function of E. coli EttA (Energy-dependent translational throttle protein), which regulates the ribosome’s entry into the translation elongation cycle, depending on the ATP/ADP ratio. We have further stated it in the first paragraph of the Main section.

6. In Discussion, the authors propose a role of Etta in A-site tRNA accommodation by displacing H89 proposed by a theoretical study (ref. 29). Cryo-EM of accommodating tRNA showed that A-tRNA accommodation is facilitated by 30S rotation (ref: <https://pubmed.ncbi.nlm.nih.gov/32612237/>). The authors therefore should mention that Etta sampling different rotations of 30S is also consistent with assisting in tRNA accommodation observed experimentally.

We thank the reviewer for the suggestion. We have added a sentence in which this cryo-EM study of experimentally observed tRNA accommodation into A-site supports our result. We have also cited this paper in our revised manuscript.